# Influence of Welding Speed on Characteristics of Non-Axisymmetric Laser-Tungsten Inert Gas Hybrid Welded Mg/Al Lap Joints with Zn Filler

**DOI:** 10.3390/ma13173789

**Published:** 2020-08-27

**Authors:** Xinze Lv, Hongyang Wang, Liming Liu

**Affiliations:** Key Laboratory of Liaoning Advanced Welding and Joining Technology, School of Materials Science and Engineering, Dalian University of Technology, Dalian 116024, China; 1661869414@mail.dlut.edu.cn (X.L.); wang-hy@dlut.edu.cn (H.W.)

**Keywords:** laser-TIG hybrid welding, Mg–Al–Zn dissimilar joint, intermetallic compounds, microstructure, fracture

## Abstract

A non-axisymmetric laser-tungsten inert gas (TIG) heat source was designed to join Mg–Al dissimilar metals with pure Zn filler at a series of welding speeds (500–900 mm/min). Laser and TIG heat sources respectively acted on Al and Mg base metals to precisely control their dissolution into the welding pool. The solidification rate of liquid metal was controlled by adjusting the welding speed, then the reaction process of Mg, Al and Zn could be accurately regulated. The results indicated that various microstructures including Al solid solution, Zn solid solution, Mg–Zn intermetallic compounds (IMCs) and eutectic structure formed in the joint produced at different speeds. Lower welding speed (500 mm/min) caused the microstructure coarsening and higher welding speed (900 mm/min) would lead to the enrichment of MgZn_2_ intermetallic compounds. At the optimal welding speed of 800 mm/min in particular, fine MgZn_2_ IMCs grains uniformly distributed in the Al and Zn solid solution. The tensile-shear load reached a maximum of 1052.5 N/cm and the joint fractured at the fusion zone near the Al base metal.

## 1. Introduction

Al alloy, as the most widely used non-ferrous metal, has a lot of attractive properties such as low density, high specific strength, excellent corrosion resistance and good formability. Mg alloy, which is the lightest structural metal, also possesses many unique features including light weight, excellent damping capacity and electromagnetic shielding performance. Both of Mg alloy and Al alloy are widely applied in automobiles, aerospace and electronic industries for weight reduction [1,2]. Obtaining high-quality Mg–Al dissimilar joints is purposeful, not only the advantages of these two metals can be fully exploit, but also the flexibility and availability of components will be improved.

The joining of Mg–Al dissimilar metals is extremely difficult, since a large amount of brittle Mg–Al intermetallic compounds (IMCs) form during the welding process, and distribute continuously in layers. In general, this difficulty can be solved by two methods, adopting solid-state bonding process or alloying the weld seam. Friction stir welding (FSW) [3,4,5,6,7,8], resistance spot welding (RSW) [9], contact-reaction brazing [10,11,12], ultrasonic spot welding [13,14,15] and explosive welding [16,17,18] were employed for joining Mg–Al dissimilar metals, and the formation of IMCs was restrained by reducing the reaction temperature and reaction time of the liquid metal. However, decreasing the heat input could not completely eliminate the brittle IMCs in the joint. Thus, some alloying elements such as Zn, Cd, Cu, Fe, Ni, Zr and Ti were added into the weld [19,20,21,22,23,24,25], the brittle Mg–Al IMCs were replaced by other compounds with low brittleness [26].

Although solid-state welding effectively inhibits the formation of brittle IMCs, they have strict requirements for the size of workpieces, welding conditions and assembly methods. Friction stir welding needs to rigidly fix the substrate sheets on the pad; Contact reaction brazing needs to be carried out in the vacuum brazing furnace; Ultrasonic welding and explosive welding need the base materials to be assembled in lap configuration. Tungsten inert gas (TIG) welding, which is a traditional fusion welding method, has been widely applied in many industries thanks to its flexible manufacturing and low cost. It also shows some potential in the joining of Mg–Al dissimilar metals. 

In our previous work, TIG welding was employed to join AZ31 Mg and 6061 Al alloy in butt configuration with Zn filler wire [27]. In this way, Mg–Al IMCs were completely eliminated and replaced by MgZn_2_ IMCs. These MgZn_2_ IMCs particles were separated by tiny amounts of plastic Al and Zn solid solution, which improved the performance of joint. However, the dense MgZn_2_ IMCs were still mainly responsible for the joint breaking due to the massive dissolution of Mg base metal. In order to accurately control the dissolution of base metals into the molten pool, a laser was introduced and formed a laser-TIG hybrid heat source. This heat source possesses better stability of the arc and higher energy density [28,29,30], which shows possibilities in Mg–Al dissimilar welding.

Because the thermal conductivity of Al (94 W·m^−1^·K^−1^) is higher than that of Mg (78 W·m^−1^·K^−1^) and the boiling point of Al (2520 °C) is more than twice than that of Mg (1090 °C), the heat input required for penetrating Al alloy is higher than that for Mg [31]. If adopting a single TIG as a heat source, a catastrophic evaporation and melting easily occurs to Mg base metal when Al alloy penetrates, leading to lots of defects and embrittlement of the joint. With regard to hybrid welding, laser and TIG in the hybrid heat source system can, respectively, act on Al and Mg base metals to precisely control their dissolution into welding pool, so as to restrain the formation of brittle MgZn_2_ IMCs in the joint. Scherm [32] reported the laser welding of Mg alloy to Al alloy by two focus optics system with Zn-Al filler, the strength of the joint was markedly improved than that by single laser by Yang [33]. Furthermore, the stability of the arc was improved due to the attraction of laser to arc [34]. Thus, the welding speed could be significantly increased in hybrid welding process, which would reduce the reaction time for liquid metals and suppress the formation of brittle IMCs.

Therefore, in this study AZ31 Mg and 6061 Al dissimilar metals were lap joined by laser-TIG hybrid welding with Zn filler wire. The relative position of the laser and TIG was designed to accurately control the distribution of heat input. The influence of welding speed on microstructures and mechanical properties was also investigated. Furthermore, the joining mechanism of the Mg–Al–Zn dissimilar joint was clarified. 

## 2. Experimental Procedure

### 2.1. Materials

We employed 6061 Al alloy sheets with a thickness of 1.5 mm and AZ31 Mg alloy sheets with a thickness of 2 mm in this study, both base metals were cut into rectangular plates with the dimension of 100 mm × 50 mm. Their nominal chemical compositions and mechanical properties are shown in Table 1 and Table 2, respectively. The filler metal is pure Zn wire with diameter of 2 mm. Before the welding process, the base metals and Zn wires were polished by sandpapers to remove the oxide film on surface.

### 2.2. Laser-Tungsten Inert Gas (TIG) Hybrid Welding Process

The heat source used in this work included a pulsed Nd:YAG laser beam and a paraxial arc generated by an OTC AEP-500P TIG welding machine. The wavelength of laser is 1.064 μm and the diameter of laser spot is 0.6 mm when being focused by a lens with a focal distance of 120 mm on the surface of the workpiece. The maximum average power of laser is 1000 W. The TIG welding machine was used in a standard AC mode to clean the oxide film on the surface of Mg and Al base metals.

The schematic illustration of non-axisymmetric laser-TIG hybrid welding process is shown in Figure 1a. The 6061 Al alloy was placed upon the AZ31 Mg alloy in lap configuration, two pure Zn wires with diameter of 2 mm were placed abreast on the Mg base metal adjacent to the lap area. In the hybrid welding process, the laser led with the arc followed behind. The angle of the arc torch and workpiece surface was 45°. The horizontal distance between laser beam and TIG electrode along welding direction (DLA) was 1.5 mm, and the distance of that perpendicular to welding direction was 2 mm. In the common laser-TIG hybrid welding process, the distance between laser and TIG perpendicular to welding direction was 0 mm to obtain a more concentrated energy density. Thus, the distribution of heat sources was axisymmetric. However, in Mg–Al dissimilar joining process, the dissolution of base metals must be restrained to avoid the formation of brittle IMCs. Therefore, laser and TIG were separated and they were not on the same plane, which is called non-axisymmetric laser-TIG hybrid welding.

In order to protect the molten pool from oxidation, pure argon with purity of 99.99% was employed as shielding gas and the flow rate was 15 L/min. A ceramic block, which did not melt during welding process, was used to fix the Zn filler. It could prevent the Zn wire from blowing away by the shielding gas and arc force. Based on the characteristics of laser and TIG, the laser beam was focused on the edge of upper Al alloy to precisely control the melting of Al base metal. The tungsten electrode was placed in the middle of two Zn wires, which was beneficial to the spread of liquid Zn filler on the surface of Mg substrate. In this work, laser power and TIG current were fixed, a series of welding speeds including 900 mm/min, 800 mm/min, 700 mm/min, 600 mm/min and 500 mm/min were adopted and the influence of welding speed on microstructures and mechanical properties was discussed in detail. The welding parameters used in the experiment are listed in Table 3.

### 2.3. Analysis Methods

The cross-sections of the joints were prepared as metallographic specimens. The samples were mechanically grinded using 400, 600, 1000 and 1500 grades of SiC sandpapers followed by polishing using a 1.5 μm diamond polishing paste. After standard grinding and polishing process, the specimens were etched by oxalic acid solution (10 g oxalic acid + 100 mL H_2_O). A metallographic microscope (DMI4000B produced by Leica, Germany) was used to observe the macrostructures of joints at different welding speeds. The microstructures of dissimilar joints were revealed by scanning electron microscope (SEM, SUPRA 55 produced by Zeiss, Germany) in secondary electron (SE2) mode. Phases in the joint were analyzed by an X-ray diffraction device (XRD, Empyrean from Malvern Panalytical, Almelo, Netherlands) using Cu target. The chemical compositions of the phases in weld seam and fracture surface were detected by energy-dispersive X-ray spectrometer (EDS). The Vickers hardness was measured from the Mg substrate to the fusion zone under a test load of 200 g and a dwell time of 10 s with an HV-1000B microhardness tester. Figure 1b shows the dimension of tensile-shear specimen. Two shims were added to ensure that the tensile direction was parallel to the axis of specimen. The tensile-shear properties were evaluated at a speed of 1 mm/min at room temperature on an electronic tension machine (Css-2205). The dimensions of tensile pieces and test parameters were referred to the study by Qi [22], and the linear loads were the average value from three specimens.

## 3. Results and Discussion

### 3.1. Design of Heat Source in Dissimilar Welding Process

Based on the characteristics of Mg–Al dissimilar joints, the heat input of laser and TIG in hybrid heat source must be reasonably allocated to control the dissolution of base metals. Single TIG, laser-TIG and non-axisymmetric laser-TIG heat sources were adopted to join Mg–Al dissimilar metals at a relatively high welding speed (800 mm/min), as shown in Figure 2.

Figure 3 shows the macroscopic morphologies of Mg–Al–Zn dissimilar joints welded by different heat sources. As indicated in Figure 3a,b, it was impossible to achieve joining between Mg and Al sheets when single TIG was used. No penetration was found on the faying surface due to the relatively low heat input. The Zn filler could not completely melt by arc and remained on the Mg substrate. 

When using laser-TIG heat source, the heat input was too concentrated, giving rise to the burn through defect, as shown in Figure 3c. In this case, most of the energy of arc was used to heat Al base metal, causing its excessive melting (marked in the black box in Figure 3d). Based on the related study in Mg/Al cold metal transfer welding [35], it would increase the possibility of forming brittle IMCs because liquid Mg and Al were easier to contact with each other. As a result, obvious cracks were found at the bottom of fusion zone (marked by the white arrows). The tensile testing samples fractured either during mechanical processing or after coupon clamping in tensile test. 

Until the non-axisymmetric laser-TIG heat source was adopted, a stable appearance without obvious defect was obtained, as shown in Figure 3e. In this case, the heat input was dispersed to avoid the excess dissolution of base metals. As a result, an acceptable melting amount of Al was observed from Figure 3f (marked in the black box). Owing to the offset of the arc, two Zn filler wires were totally melted into the weld pool. This could reduce the dilution ratio of Mg and Al base metals in dissimilar joints and inhibit the formation of brittle Mg–Al IMCs. No crack was found in the cross section of the joint from Figure 3f. Therefore, a non-axisymmetric laser-TIG heat source is more suitable for joining Mg–Al dissimilar metals. All of the following joints were produced by this heat source.

### 3.2. Macrostructures and Properties of Joints

From our previous study, the weak area of the Mg–Al–Zn butt joint made by TIG is the fusion zone near Mg alloy base material. Fracture occurred at the interface between the MgZn_2_ particles and (Al, Zn) solid solution [27], as shown in Figure 4. Through the control of welding speed, it is possible to obtain a joint with more (Al, Zn) solid solution and less MgZn_2_ IMCs by increasing the solidification rate of welding pool. 

The macroscopic morphologies of Mg–Al–Zn dissimilar joints produced at different welding speeds are shown in Figure 5. When the welding speed was larger than 900 mm/min, it was hard to obtain a continuous joint because liquid Zn filler spread poorly on Mg substrate. As the welding speed decreased, acceptable joints could be obtained as the weld penetration and width in the Mg substrate increased. Meanwhile, some porosities gradually formed and grow up in the weld seam due to the excessive vaporization of Zn filler. Note that an obvious stratification phenomenon appeared, especially in the case of high welding speed, which resulted from the inhomogeneous microstructures in rapid solidification process. Thus, the joint is divided into three typical areas for following analysis: Part I is fusion zone (FZ). It includes two parts, one is mainly composed of the melting Zn filler (FZ_Zn_) and the other is the mixture zone of liquid Zn and Mg (FZ_Mg_); Part II is the Al/FZ interface; Part III is the Mg/FZ interface, as shown in Figure 6. 

Figure 7a shows the tensile-shear load as a function of welding speed. As could be seen, the tensile-shear load gradually rose with welding speed increased from 500 mm/min to 800 mm/min and reached a maximum value of 1052.5 N/cm, then slightly dropped to 986.5 N/cm with further increasing welding speed. The typical force-displacement curves are shown in Figure 7b. A yielding phenomenon (marked in the black box in Figure 7b) was found in the joint with highest fracture load (made at 800 mm/min). We inferred that plastic deformation occurred in FZ_Zn_ because the dilution rate in this region was the lowest. The weld metal probably consisted of Zn filler and slight plastic deformation occurred before fracturing. Further analysis could be seen in the fracture mechanism in Section 3.6. 

After tensile-shear strength tests, the fractured joints were re-combined to observe the fracture locations, the results were shown in Figure 8. For the joint welded at 900 mm/min, the fracture occurred in FZ_Mg_ near Mg substrate. As the welding speed decreased, all the joints fractured in FZ near Al base metal. The fracture started at FZ_Mg_ and propagated to FZ_Zn_, just as shown in Figure 8b–d. Seen from Figure 8c, the fracture had passed through the porosity, indicating that the relatively low welding speed would cause the porosity defect and deteriorate the joint performance. With the help of fracture locations, FZ_Zn_ and FZ_Mg_ were identified as the weak areas. Thus, the microstructures in these areas were investigated emphatically.

### 3.3. Microstructures

The phase constituents of the fusion zones at different welding speeds were examined by XRD detection as shown in Figure 9. It could be seen that there were no obvious differences of the phase composition among the joints at different welding speeds. The fusion zones mainly consisted of Zn solid solution and MgZn_2_ compounds. The Al and Mg solid solution were also detected as base materials. However, whether they exist in the weld seam still needs to be further confirmed by EDS analysis.

Welding speed affects the content of Mg, Al and Zn in liquid metals, which has a main influence on the microstructural evolution. Figure 10 shows the SEM micrographs of Al/FZ interface at different welding speeds. For the joints made at 900–700 mm/min, some dendrite crystals were found to grow from Al base metal perpendicularly into FZ. Based on the EDS results listed in Table 4, the dendrites (phase-1A, phase-1D and phase-1F) were identified as Al solid solution; while the phases near these Al dendrites (phase-1B, phase-1C and phase-1E) were confirmed as the mixture of Zn solid solution and Al solid solution (MZAS). Based on the Al–Zn binary diagram, the maximum solid solubility of Zn in Al is 67 at.% at 381 °C, thus Zn solid solution will precipitate from the supersaturated Al grains and form MZAS.

As the welding speed decreased (700–500 mm/min), some phases containing Mg (phase-1G and phase-1I) were found near the Al/FZ interface as more Mg atoms diffused into molten pool. Based on the EDS results in Table 4, they were identified as MZAS + MgZn_2_ eutectic and MgZn_2_, respectively. For the joints welded at 500 mm/min, the Al dendrite at the interface completely disappeared, suggesting that its growth mode transformed from dendritic growth to planar growth, which was determined by the low temperature gradient at relatively low welding speed.

Figure 11 shows the microstructures of FZ_Zn_ at different welding speeds. For the joint welded at 900 mm/min, it was composed of some equiaxed grains (phase-2A) with size of 10–15 μm and the striped phases (phase-2B) distributed at the grain boundaries. Based on the EDS analysis results in Table 5, they were the Zn solid solution and MZAS, respectively. Furthermore, the MZAS (phase-2B) was identified as the eutectic of Al solid solution and Zn solid solution. It was noticed that the Zn grains could not close contact and the MZAS distributing at the boundary was not continuous. In rapid solidification process, the eutectic liquid formed “liquid films” along the grain boundaries, could not fill in the space caused by the solidification shrinkage, leading to the grain boundary gully, as shown in Figure 11a. 

As the welding speed further decreased to 700 mm/min, some phases with lamellar structure (phase-2E and phase-2H) were found near Zn and Al grains. They exhibited a typical characteristic of eutectic structure as shown in the inset in Figure 11b. The EDS results indicated that the structure was MZAS + MgZn_2_ eutectic. A similar eutectic structure was also found in the Mg–Al butt joints welded by TIG with Zn-30Al filler [36]. The number of eutectic depended on the Mg content in liquid, which was determined by the heat input. As a result, the amount of eutectic structure increased distinctly with the welding speed decreased, as shown in Figure 11b,c. Meanwhile the Zn grains were gradually refined, suggesting partial Zn entered the eutectic structure and formed MgZn_2_ IMCs. 

The microstructure of joint produced at 500 mm/min is shown in Figure 11d. The dark grey polygynous phases (phase-2I) were surrounded by the light grey irregular phases (phase-2J). Based on the energy-dispersive X-ray spectroscopy (EDS) results in Table 5, they were confirmed as MgZn_2_ and MZAS, respectively. The high heat input promoted the dissolution of Mg base metal, resulting in the formation of Mg-rich phase MgZn_2_ in large size. 

Figure 12 shows the microstructures of FZ_Mg_ at different welding speeds, they exhibited the similar microstructures with dark grey particles dispersed into the light grey matrix. In FZ_Mg_, it is possible to occur the following reactions based on the Mg–Zn binary diagram:L→MgZn_2_ High temperatureL + MgZn_2_→Mg_2_Zn_11_ 381 °CL→Mg_2_Zn_11_ + (Zn) 364 °CL→MgZn_2_ + (Zn) 364 °C

However, no Mg_2_Zn_11_ compounds were detected based on the XRD result shown in Figure 9. The fast cooling rate inhibited the peritectic reaction and the formation of Mg_2_Zn_11_ compounds. According to the EDS results shown in Table 6, the dark grey particle (phase-3A) was composed of 20.83 at.% Mg, 6.53 at.% Al and 72.64 at.% Zn, which was identified as MgZn_2_ compounds. The light grey matrix (phase-3B), whose content of Mg significantly dropped, was confirmed as (Zn) + little MgZn_2_. Similar observation results also appeared in the ultrasonic-assisted semi-solid brazing on dissimilar Al-Mg alloys [37]. At the extremely fast cooling rate of non-axisymmetric laser-TIG welding, the reaction of Mg and Zn was restrained. As a result, the MgZn_2_ IMC particles were visibly refined and dispersed compared with that by single TIG in Figure 4b. The weak area of the joint also changed from FZ_Mg_ (by single TIG) to FZ_Zn_ (by non-axisymmetric laser-TIG).

As the decrease of welding speed, the content of Mg and Al in liquid metal increased. Therefore, the light grey matrix (phase-3D, phase-3F and phase-3H) was prone to excrete Mg and form Al-rich MZAS according to the EDS results in Table 6. 

Despite the similar microstructural characteristics, differences still can be observed from the distribution of IMCs. For the joint welded at 900 mm/min, the dark grey MgZn_2_ particles were found to be enriched in some region (marked in the black circles), which was called the “IMCs-rich zone”. As the welding speed decreased, the IMCs-rich zones disappeared and the MgZn_2_ IMCs became more dispersed due to the relatively sufficient Mg, Al and Zn liquid mixing. However, extremely low welding speed (500 mm/min) would cause the microstructure coarsening, just like the large MgZn_2_ grains shown in Figure 12d. The distribution of MgZn_2_ IMCs had a great influence on the joint performance, which can be seen in fracture mechanism in Section 3.6. 

Figure 13 shows the microstructures of the Mg/FZ interface. A transitional layer was found between the Mg base metal and FZ (marked by the white dotted lines). The thickness of this layer increased distinctly with the decrease of welding speed. For the joint welded at 900 mm/min, the transitional layer (phase-4A) was identified as Mg solid solution + MgZn since it contained 58.42 at.% Mg, 6.08 at.% Al and 35.50 at.% Zn. The MgZn compound was not found by the XRD detection because its content was too low. As the welding speed decreased, a lamellar structure newly appeared in the transitional layer. Base on the EDS results in Table 7, it was confirmed as Mg solid solution + MgZn eutectic (phase-4B, phase-4D and phase-4F), which corresponded to the observation results when using diffusion bonding of Mg to Al with Zn interlayer [38]. Note that some polygonal particles (phase-4E), which had the similar composition with phase-4A, were dispersed in the matrix of eutectic layer. They were also confirmed as Mg solid solution + MgZn. The formation of this phase was determined by the composition fluctuation of the eutectic liquid, which changed it into dispersed particles after solidification. These small particles would not deteriorate the property of joint, due to their dispersed distribution characteristic according to the related studies by Tan [39]. 

### 3.4. Microhardness Distribution

Figure 14 shows the micro-hardness distribution of joints welded at different speeds. The hardness is closely associated with the plasticity of joint, which has a major influence on the mechanical properties of dissimilar metals. As could be seen, the hardness quickly increased from the Mg substrate to the Mg/FZ interface, which resulted from the formation of Mg–Zn IMCs. Afterwards the hardness gradually decreased from FZ_Mg_ to FZ_Zn_ due to the limited diffusion range of Mg atoms. As the welding speed decreased from 900 mm/min to 500 mm/min, more MgZn_2_ IMCs and eutectic structure formed, giving rise to the increase of hardness in FZ_Zn_. An interesting phenomenon was that the hardness at Mg/FZ interface abnormally dropped with welding speed decreased from 900 mm/min to 700 mm/min. It was probably related to the elimination of IMC-rich zones. The more dispersed MgZn_2_ IMCs and uniform microstructure resulted in the decrease of hardness.

### 3.5. Formation of Phases and Solidification Process

Based on the microstructural analysis above, the solidification process was expected to be clarified with the assistance of the schematic diagram shown in Figure 15. At the Al/FZ interface, the upper Al base metal melted under the irradiation of laser and diffused into the Zn liquid (Figure 15a). As the temperature dropped to 560 °C, Al dendrites precipitated primarily at the solid–liquid interface and grew into FZ (Figure 15b). When the temperature further decreased to 490 °C, Al-rich MZAS formed at the tip of dendrite (Figure 15c). Finally as the temperature reached 430 °C, Zn-rich MZAS precipitated from the supersaturated Al dendrites and remaining liquid (Figure 15d). No Mg–Zn IMCs formed in this region due to the limited diffusion of Mg atoms at relatively high welding speed. 

According to Figure 15e, a small amount of Mg atoms had diffused to FZ_Zn_. As the temperature dropped to 550 °C, Al grains first precipitated from the liquid (Figure 15f). Subsequently based on the Mg–Al–Zn ternary diagram shown in Figure 16, the composition of liquid reached e_2_E_2_ line and the eutectic reaction occurred: L→(Al) + MgZn_2_ at 490 °C (Figure 15g). Finally, a large amount of Zn grains formed based on the reaction: L→(Al) + MgZn_2_ + (Zn) at 400 °C (Figure 15h). 

Under the heating of the arc, melting and interdiffusion of Mg and Zn took place in FZ_Mg_ (Figure 15i). Fine MgZn_2_ IMCs particles first precipitated from the liquid at 590 °C. At the same time, a layer of MgZn_2_ was deposited at the liquid-solid interface. As the consumption of Mg in liquid, MZAS began to form along the MgZn_2_ IMCs boundaries at 460 °C (Figure 15k). Finally, a eutectic reaction between Mg and Zn occurred at the liquid–solid interface at 340 °C, giving rise to the (Mg) + MgZn eutectic (Figure 15l). The composition fluctuation of Mg in liquid caused the variation of precipitated phases in the solidification process.

### 3.6. Microstructure Evolution and Fracture Mechanism

Figure 17 shows the schematic of microstructure evolution at different welding speeds. The distributions of brittle IMCs in weak areas (FZ_Mg_ and FZ_Zn_) were mainly discussed, and the relationship between microstructures and mechanical properties was established. 

As the welding speed decreased from 900 mm/min to 800 mm/min, the tensile-shear load of the joint increased with the fracture location changing from FZ_Mg_ to FZ near Al base metal, indicating that the property of FZ_Mg_ was improved. Based on the microstructure evolution in Figure 17, the brittle IMCs particles in FZ_Mg_ became more dispersed and completely separated by the MZAS phases as the welding speed decreased. The elimination of IMC-rich zones was attributed to the proper solidification rate, which not only ensured the sufficient mixing of liquid Mg and Zn, but also avoided the excessive dissolution of Mg base metal. The plastic MZAS phases could impede the stress concentration caused by brittle MgZn_2_ IMCs and reduce the possibility of crack propagation, resulting in the increase of strength. 

When further decreasing the welding speed to 700 mm/min, the amount of eutectic in the weak area (FZ_Zn_) increased significantly, along with the Zn grains gradually refined. Eventually, at the lowest welding speed of 500 mm/min, FZ_Zn_ formed primarily MgZn_2_ IMCs rather than eutectic structure or Zn solid solution. Based on the previous analysis, the eutectic structure is the mixture of MZAS and MgZn_2_ IMCs, and its plasticity is superior to brittle IMCs but inferior to Al and Zn solid solution. Thus as the welding speed decreased, the joints became brittle gradually, which caused the mechanical performance to deteriorate. 

The fracture surface morphologies and corresponding micro-fracture locations are presented in Figure 18. For the joint welded at 900 mm/min, it was found to exhibit the characteristics of cleavage fracture. Based on the micro-fracture location in Figure 18b, the fracture occurred along the MgZn_2_ IMCs particles and propagated to the MZAS matrix in FZ_Mg_. The dense MgZn_2_ IMCs were the main crack source due to their intrinsic brittleness.

The joint welded at 800 mm/min displayed a typical quasi-cleavage fracture feature, which usually appeared in HCP (hexagonal close-packed) and BCC (body-centered cubic) structure metals. The micro-fracture location showed that breaking occurred along the Zn grain boundaries. Note that some shallow dimples were found in the fracture surface (pointed by the white arrows), indicating slight deformation occurred before fracture and giving rise to the highest tensile-shear load. 

For the joint welded at 700 mm/min, the fracture surface was characterized by plenty of stripped tear edges, which corresponded to the eutectic structure observed in the fracture location in Figure 18f. In this case, the number of plastic Zn grains was reduced compared with that at 800 mm/min. Moreover, some porosities were also found on the fracture surface (pointed by the white arrows), which deteriorated the joint performance. The joint welded at 500 mm/min displayed a typical brittle fracture feature with many smooth surfaces without adequate plastic deformation. The MgZn_2_ IMCs with a large size were mainly responsible for the joint breaking. 

## 4. Conclusions

Non-axisymmetric laser-TIG hybrid welding of Mg alloy to Al alloy with Zn filler was performed, and the influence of welding speed on microstructures and mechanical properties of the Mg–Al–Zn dissimilar joint was clarified. The main conclusions of this research can be summarized as follows:(1)The non-axisymmetric laser-TIG heat source could reduce the dilution ratio of Mg and Al base metals in the weld seam, thus the joint embrittlement happening in common laser-TIG welding processes was avoided by use of this heat source.(2)The microstructure could be regulated by changing the welding speed. For the joint made at 900 mm/min, the MgZn_2_ IMCs in FZ_Mg_ were found to be locally enriched. At 500 mm/min welding speed, a considerable number of MgZn_2_ IMCs with large size appeared and resulted in the joint embrittlement. In the case of optimal welding speed of 800 mm/min, a proper amount of MgZn_2_ IMCs and a uniform microstructure formed in the weld seam, giving rise to the highest tensile-shear strength.(3)Compared with the joint made by TIG, the brittle MgZn_2_ IMCs in the non-axisymmetric laser-TIG hybrid joint were significantly refined, and the amount of plastic MZAS phases increased. As a result, the hybrid joint broke at FZ_Zn_ rather than FZ_Mg_ and the tensile-shear load of the joints made at 800 mm/min reached a maximum value of 1052.5 N/cm.

## Figures and Tables

**Figure 1 materials-13-03789-f001:**
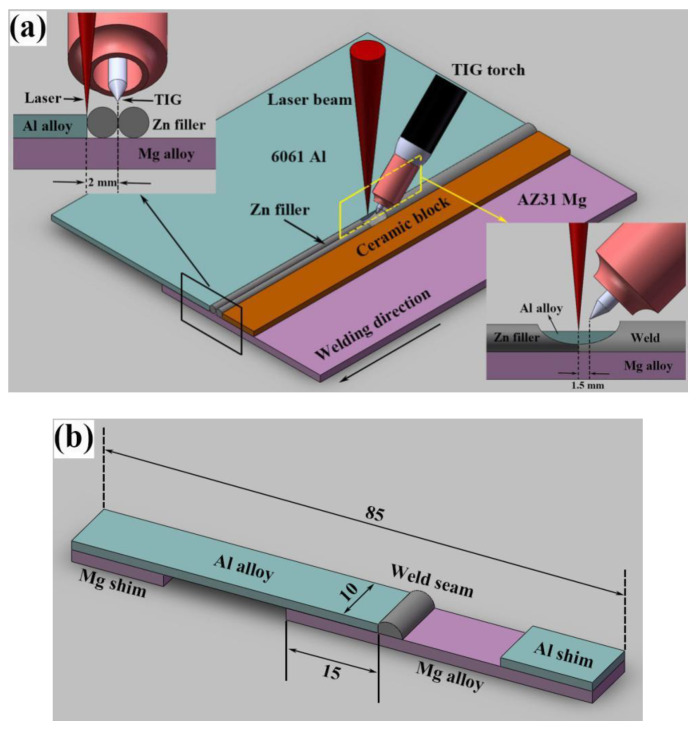
Schematic illustration of non-axisymmetric laser-tungsten inert gas (TIG) hybrid welding process and testing specimen: (**a**) hybrid welding process; (**b**) testing specimen (mm).

**Figure 2 materials-13-03789-f002:**
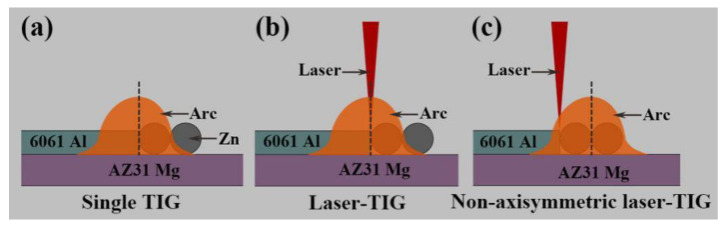
Schematic illustration of different heat sources: (**a**) Single TIG; (**b**) Laser-TIG; (**c**) Non-axisymmetric laser-TIG.

**Figure 3 materials-13-03789-f003:**
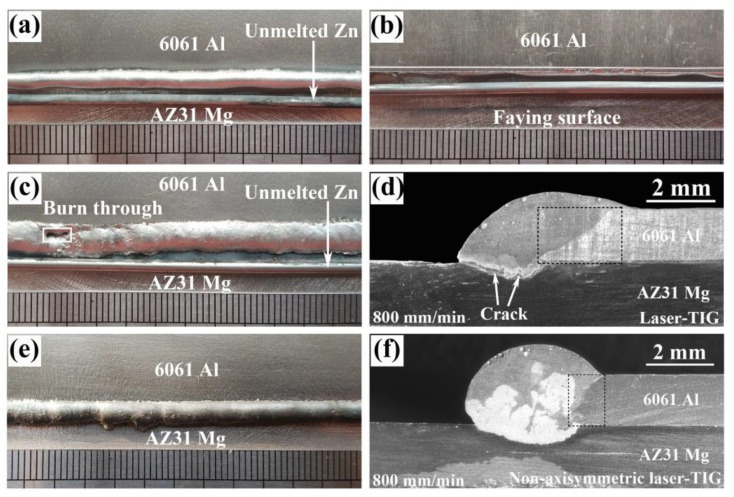
Macroscopic morphologies of Mg–Al–Zn dissimilar joints made by different heat sources: (**a**) single TIG, (**b**) faying surface of (**a**,**c**,**d**) Laser-TIG, (**e**,**f**) Non-axisymmetric laser-TIG.

**Figure 4 materials-13-03789-f004:**
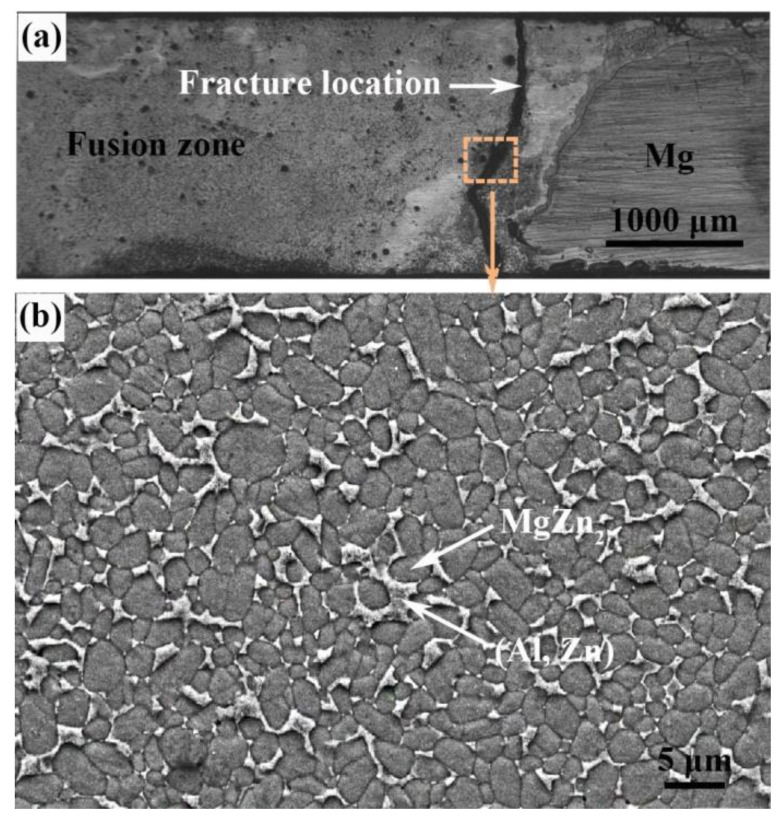
(**a**) Macro-fracture location of Mg–Al–Zn butt joint with TIG, (**b**) microstructure of the weak area in (**a**).

**Figure 5 materials-13-03789-f005:**
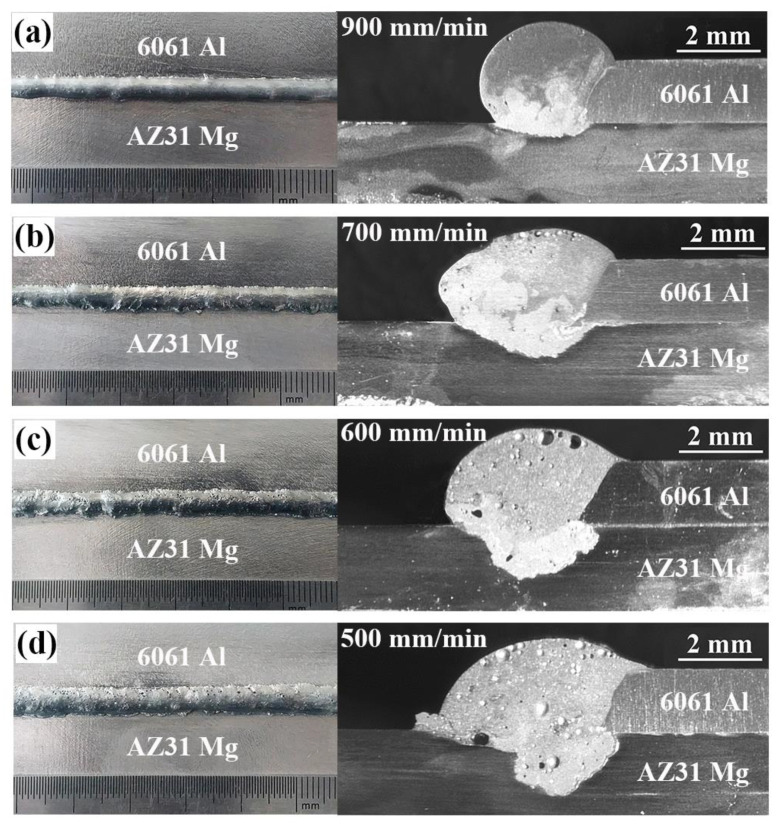
Appearances and corresponding cross sections of non-axisymmetric laser-TIG welded Mg–Al–Zn dissimilar joints at different welding speeds: (**a**) 900 mm/min, (**b**) 700 mm/min, (**c**) 600 mm/min, (**d**) 500 mm/min.

**Figure 6 materials-13-03789-f006:**
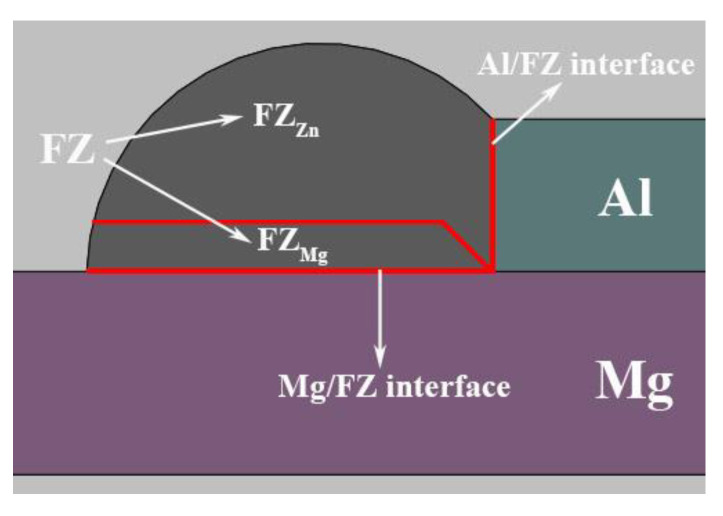
Schematic illustration of typical areas in Mg–Al–Zn dissimilar joints.

**Figure 7 materials-13-03789-f007:**
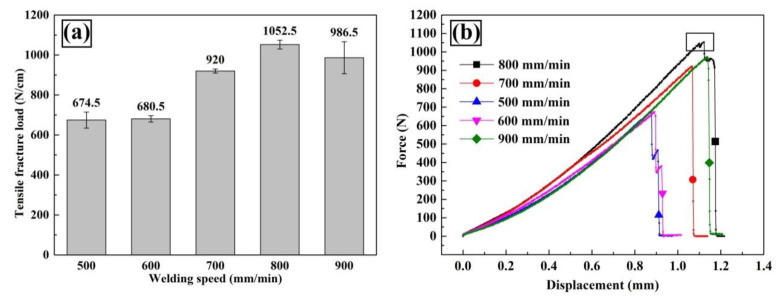
Mechanical properties of Mg–Al–Zn dissimilar joints made at different welding speeds: (**a**) tensile-shear loads, (**b**) force-displacement curves.

**Figure 8 materials-13-03789-f008:**
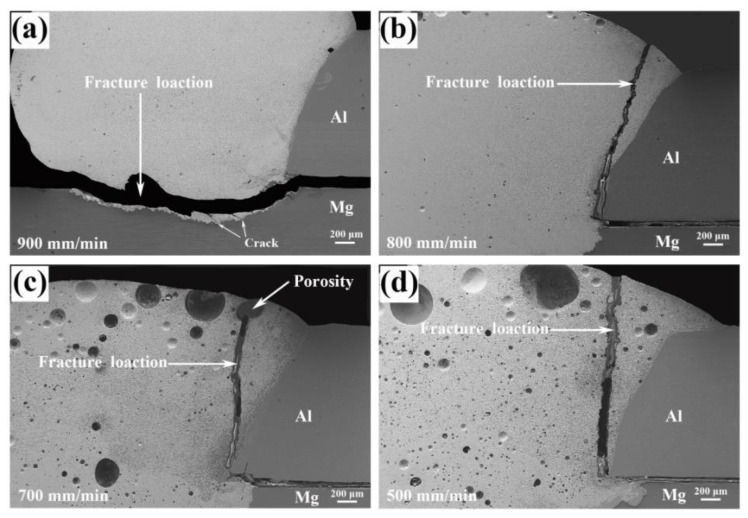
Macro-fracture locations of Mg–Al–Zn dissimilar joints at different welding speeds: (**a**) 900 mm/min, (**b**) 800 mm/min, (**c**) 700 mm/min, (**d**) 500 mm/min.

**Figure 9 materials-13-03789-f009:**
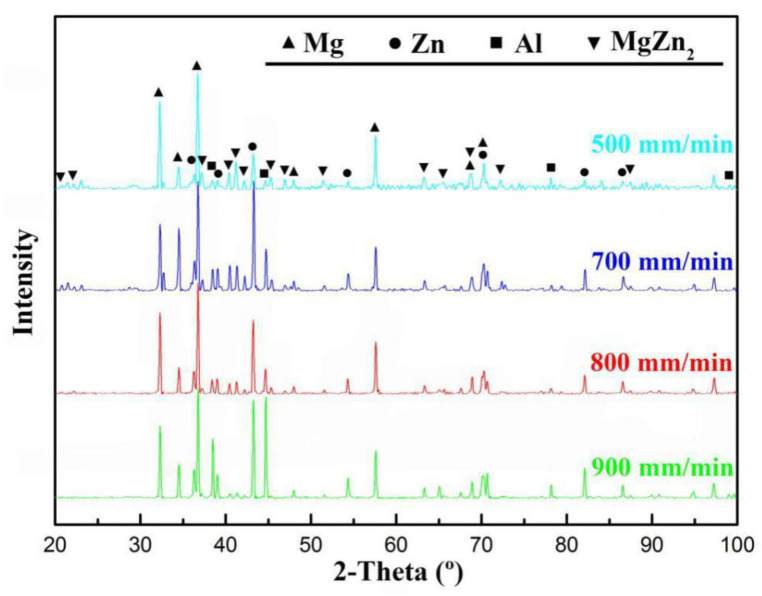
X-ray diffraction (XRD) patterns of phases in fusion zone (FZ) at different welding speeds.

**Figure 10 materials-13-03789-f010:**
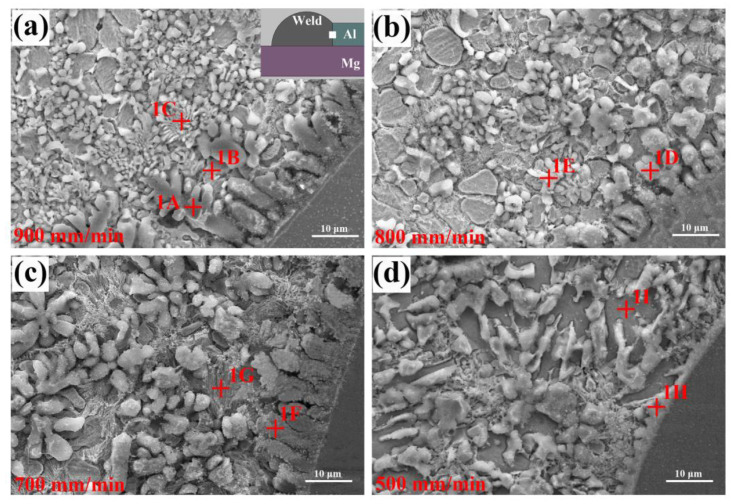
Microstructures of Al/FZ interface at different welding speeds: (**a**) 900 mm/min, (**b**) 800 mm/min, (**c**) 700 mm/min, (**d**) 500 mm/min.

**Figure 11 materials-13-03789-f011:**
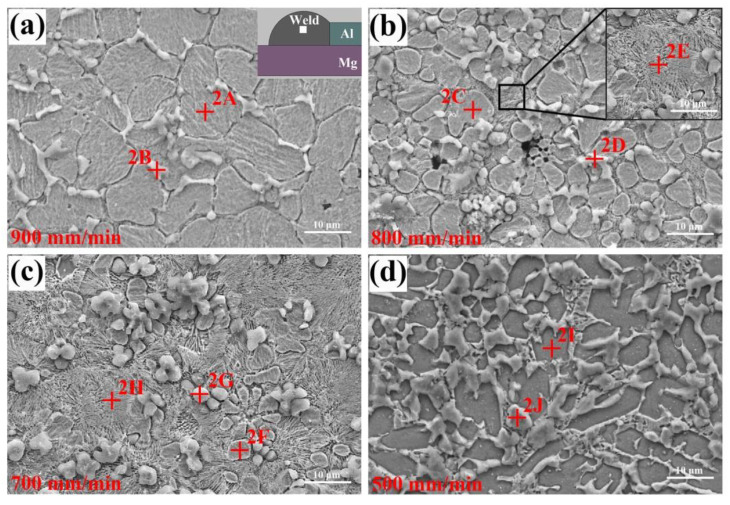
Microstructures of FZ_Zn_ at different welding speeds: (**a**) 900 mm/min, (**b**) 800 mm/min, (**c**) 700 mm/min, (**d**) 500 mm/min.

**Figure 12 materials-13-03789-f012:**
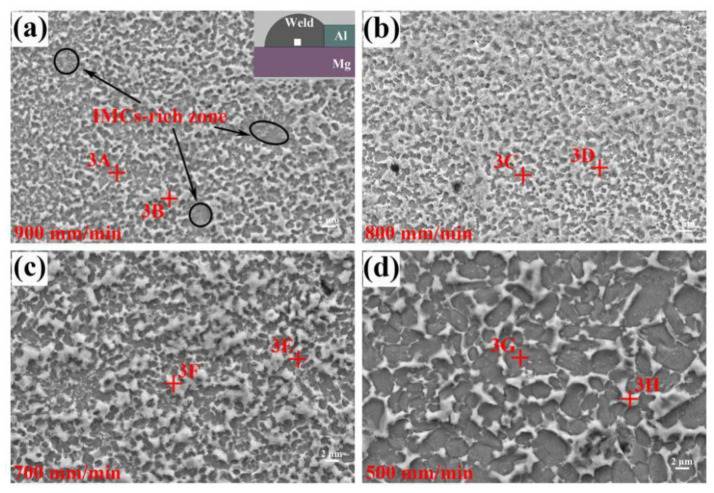
Microstructures of FZ_Mg_ at different welding speeds: (**a**) 900 mm/min, (**b**) 800 mm/min, (**c**) 700 mm/min, (**d**) 500 mm/min.

**Figure 13 materials-13-03789-f013:**
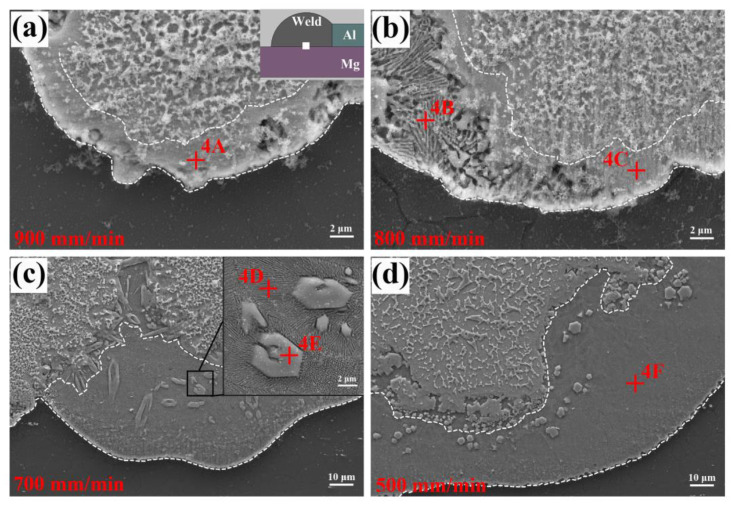
Microstructures of Mg/FZ interface at different welding speeds: (**a**) 900 mm/min, (**b**) 800 mm/min, (**c**) 700 mm/min, (**d**) 500 mm/min.

**Figure 14 materials-13-03789-f014:**
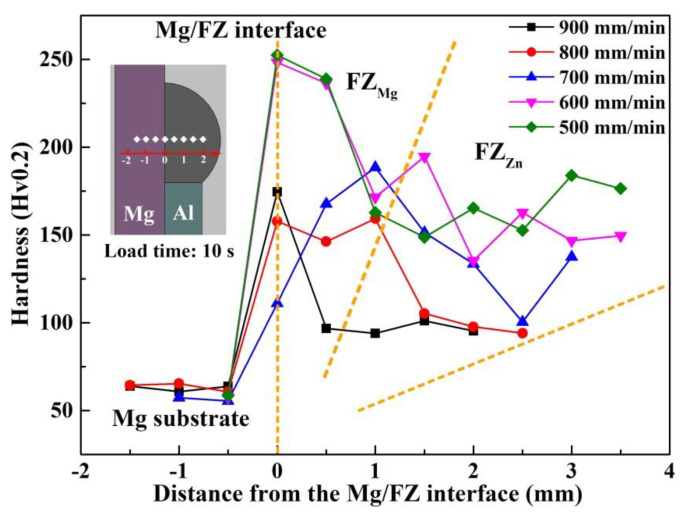
Hardness distribution of joints at different welding speeds.

**Figure 15 materials-13-03789-f015:**
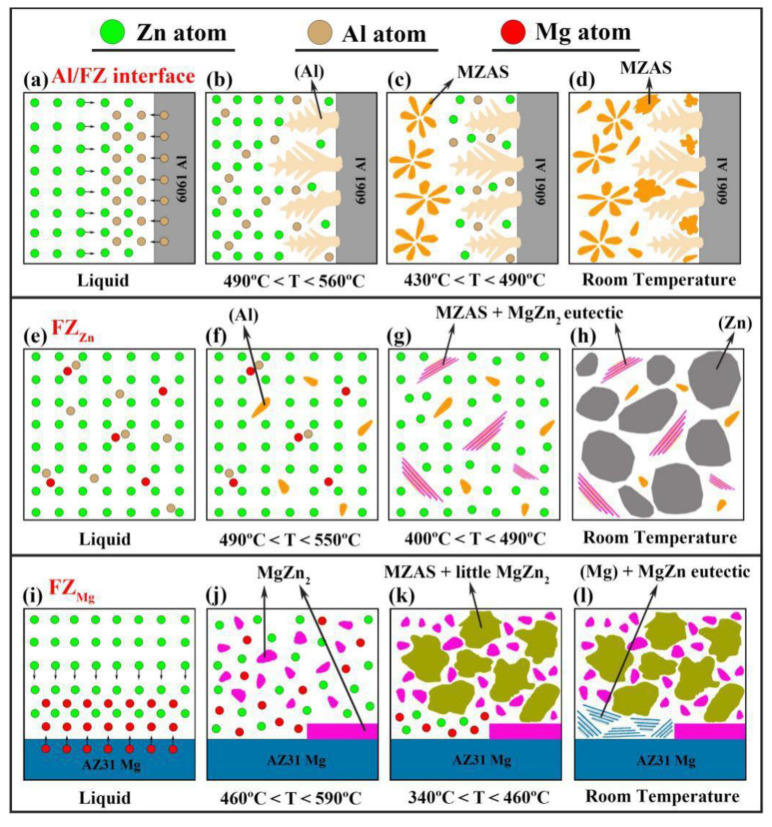
Schematic of solidification process of the joint welded at 800 mm/min: (**a**–**d**) Al/FZ interface; (**e**–**h**) FZ_Zn_; (**i**–**l**) FZ_Mg_.

**Figure 16 materials-13-03789-f016:**
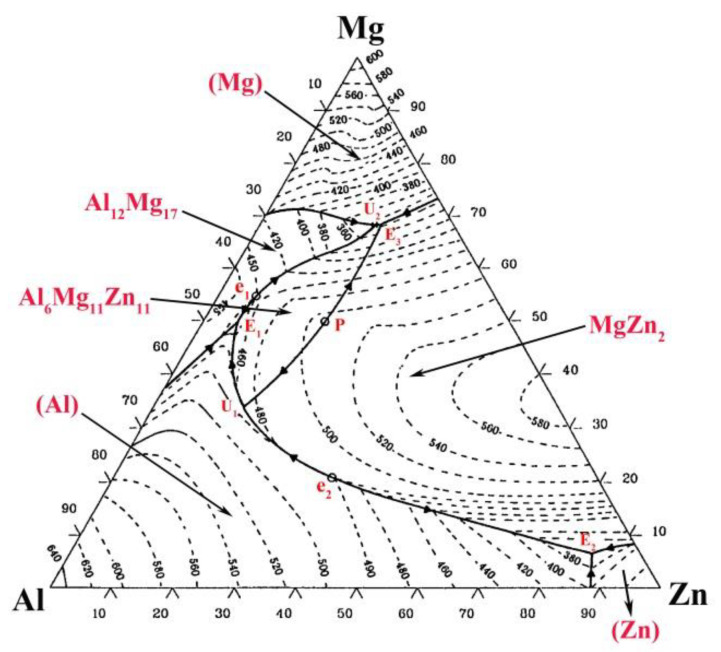
Mg–Al–Zn ternary diagram.

**Figure 17 materials-13-03789-f017:**
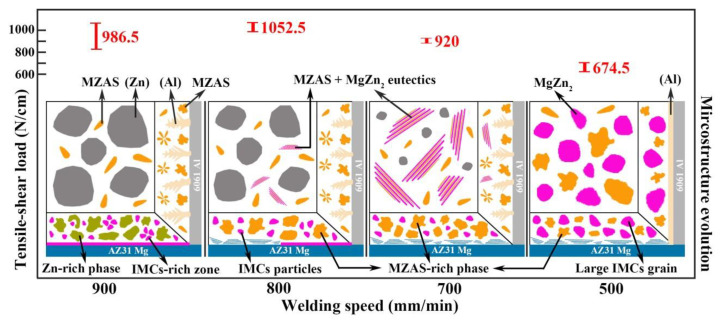
Schematic of microstructure evolution at different welding speeds.

**Figure 18 materials-13-03789-f018:**
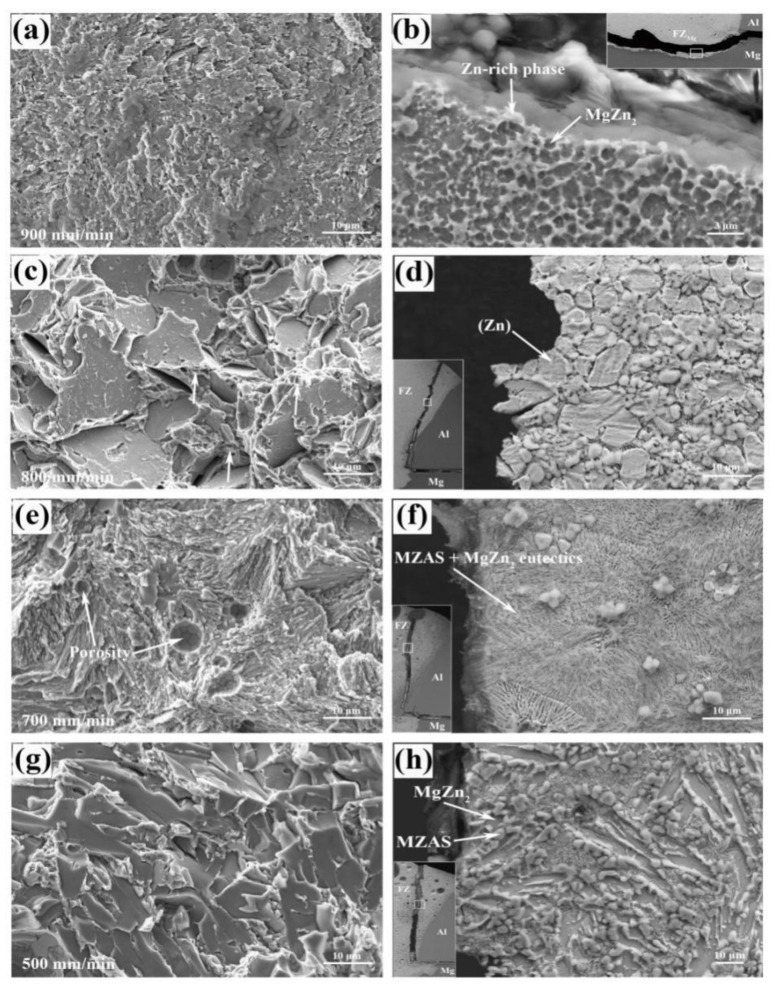
Fracture surface morphologies and corresponding micro-fracture locations at different welding speeds: (**a**,**b**) 900 mm/min, (**c**,**d**) 800 mm/min, (**e**,**f**) 700 mm/min, (**g**,**h**) 500 mm/min.

**Table 1 materials-13-03789-t001:** Chemical compositions of 6061 Al alloy and AZ31 Mg alloy (wt.%).

Materials	Al	Mg	Si	Zn	Cu	Mn
6061 Al Alloy	Bal.	1.00	0.60	-	0.15	0.01
AZ31 Mg Alloy	3.00	Bal.	0.10	1.00	-	0.20

**Table 2 materials-13-03789-t002:** Mechanical properties of 6061 Al alloy and AZ31 Mg alloy.

Materials	Yield Strength (MPa)	Tensile Strength (MPa)	Shear Strength (MPa)	Elongation (%)
6061 Al Alloy	255	290	186	12
AZ31 Mg Alloy	200	260	120	10

**Table 3 materials-13-03789-t003:** Welding parameters used in the experiment.

Welding Parameters	Value
Laser Power (W)	420
Laser Pulse Duration (ms)	3
Laser Pulse Frequency (Hz)	30
Laser Defocus Distance from Al Surface (mm)	0
TIG Current (A)	80
Arc Length from Mg Surface (mm)	2
Welding Speed (mm/min)	500–900

**Table 4 materials-13-03789-t004:** Energy-dispersive X-ray spectrometer (EDS) analysis results of the phases in Figure 10 (at.%).

Position	Mg	Al	Zn	Possible Phase
1A	-	82.91	17.09	(Al)
1B	-	55.40	44.60	MZAS
1C	-	49.94	50.06	MZAS
1D	0.7	70.33	28.97	(Al)
1E	-	45.72	54.28	MZAS
1F	-	75.68	24.32	(Al)
1G	9.94	17.78	72.29	MZAS + MgZn_2_ eutectic
1H	-	65.71	34.29	(Al)
1I	30.62	7.99	61.39	MgZn_2_

**Table 5 materials-13-03789-t005:** EDS analysis results of the phases in Figure 11 (at.%).

Position	Mg	Al	Zn	Possible Phase
2A	-	4.32	95.68	(Zn)
2B	1.16	11.09	87.75	MZAS
2C	-	7.01	92.99	(Zn)
2D	-	66.67	33.33	(Al)
2E	3.15	44.25	52.60	MZAS + MgZn_2_ eutectic
2F	-	4.48	95.52	(Zn)
2G	-	62.61	37.39	(Al)
2H	10.58	14.08	75.33	MZAS + MgZn_2_ eutectic
2I	33.13	4.73	62.13	MgZn_2_
2J	1.99	62.01	36.00	MZAS

**Table 6 materials-13-03789-t006:** EDS analysis results of the phases in Figure 12 (at.%).

Position	Mg	Al	Zn	Possible Phase
3A	20.83	6.53	72.64	MgZn_2_
3B	11.08	4.37	84.55	(Zn) + little MgZn_2_
3C	24.41	6.56	69.03	MgZn_2_
3D	6.88	35.94	57.18	MZAS + little MgZn_2_
3E	32.56	4.50	62.94	MgZn_2_
3F	9.18	52.13	38.69	MZAS + little MgZn_2_
3G	33.45	6.54	60.01	MgZn_2_
3H	1.22	52.79	45.99	MZAS

**Table 7 materials-13-03789-t007:** EDS analysis results of the phases in Figure 13 (at.%).

Position	Mg	Al	Zn	Possible Phase
4A	58.42	6.08	35.50	(Mg) + MgZn
4B	64.32	5.84	29.84	(Mg) + MgZn eutectic
4C	33.27	2.99	63.74	MgZn_2_
4D	67.80	5.29	26.91	(Mg) + MgZn eutectic
4E	51.63	7.44	40.93	(Mg) + MgZn
4F	65.12	9.19	25.69	(Mg) + MgZn eutectic

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
