# Peer review of "Influence of Welding Speed on Characteristics of Non-Axisymmetric Laser-Tungsten Inert Gas Hybrid Welded Mg/Al Lap Joints with Zn Filler"

_materials, 2020, doi:10.3390/ma13173789_

Round 1

Reviewer 1 Report

In this paper, the authors investigate the structure and mechanical properties of non-axisymmetric hybrid welded Mg/Al lap joints with Zn filler. The results provide a useful contribution to the new welding technology for Mg-Al dissimilar combination. Accordingly, the subject matter is deemed suitable for publication of Materials. However, the authors should address the flowing points prior to publication of this manuscript

1. First of all, there are several mistakes in English. The manuscript should be reviewed carefully.

2. The main variable in this study is “welding speed”, but its importance is not mentioned in Introduction. The authors should provide the reason why the “welding speed” is important in this study.

3. Please explain the non-axisymmetric welding more in detail.

4. In Fig. 7(b), yielding is not clear. Where does the plastic deformation occur, Mg, Al or weld metal?

5. In Fig. 8, the crack initiation sites seem to be same in all specimens. In this case, the crack initiation is more important than the crack propagation. So the discussion on the crack initiation should be provided with the experimental observation.

6. Concerning Table 3, 4, 5, and 6, the exact phase cannot be determined only by EDS results. The clearer evidences for phase identification should be provided, such as electron diffraction patterns in TEM.

Reviewer 2 Report

Dear authors thank You for the interesting paper. In my opinion it deserves to be published, but it needs to be improved in order to be accepted for publication. The title corresponds to the content presented in the paper. The used method is not relatively new so good that the authors focused on the speed of the process but it needs more discussion. Some points which the authors should focus on:

  • The introduction is too poor, the authors must extend the literature review as it seems it didn’t take many recent publications on this topic. As we analyze the references in this paper we can notice that over 95% of them are from Scientists from the Asia region. For a potential reader it would seem that only in this area of the world we can find scientists which are working on this topic. It does not look suitable for a high class journal like Materials as the authors have performed a poor review in terms of world renown references. Therefore the authors should analyze the literature outside of the Asia region.
  • More information about the mechanical testing procedure should be added such as the used test stand. Other mechanical properties of the materials should be added in a table.
  • More information about the benefits of using this method in comparison to others such as FSW or explosive welding should be added.
  • The tables should be reorganized so that a table is not on two different pages.
  • The structure of the whole paper is messy as we can notice by the position of the points describing the main parts of the paper.
  • The authors should omit statements such as: “As everyone knows..” without pointing literature references.
  • The authors should correct the citing references in brackets as it has a different size than the text and is not according to the citation type for mdpi journals.
  • The conclusions should be more concise.

Reviewer 3 Report

Welding of Al/Mg alloy is still problematic issue, so the topic is actual. The manuscript is well written in good English and the sections are logically divided. The study can be interesting for readers involved in welding of lightweight alloys. However, some corrections and/or amendments are suggested:

L 71: What is the roughness of the sandpaper?

L 85: “The horizontal distance between laser beam and TIG electrode along welding direction (DLA) was kept 1.5 mm.“ But in the Fig. 1 a is the horizontal distance 2 mm! Which statement is correct?

Fig. 13: It would be nice to have microhardness measurements also between Al-alloy substrate and Zn filler. The fracture was located there in most cases.

Tables 3 -5: Possible phases and their mixtures are assigned based on SEM/EDS results only. This is main issue of this manuscript. EDS is rather estimation than identification. I can not agree with the statements in the Results and Discussion that stoichiometric phases were “identified” or “confirmed” by this technique. Proper XRD or TEM electron diffraction is required for such claims.

Reviewer 4 Report

The article is interesting, it brings knowledge about the use of hybrid - laser - TIG welding technology in the case of welding dissimilar joints. The English language is at a good level and the sentences are understandable.
This is an interesting scientific article for welding engineers.

While reading, I found a few things that could be improved:
1.Please add Argon purity (eg 99.999% purity is called Ar 5.0).
2. Was the shielding gas supplied only through the nozzle of the TIG torch or as shielding gas with the nozzle directed away from the laser head?
3. Table 1 shows the chemical composition of the materials used in the experiment. Please add how these data are obtained - I suppose this is an EDS analysis.
4th line 76 - "pulse Nd: YAG laser" since a pulsed laser is used, one of the parameters should be defined as pulse length or frequency. There is no such information in Table 2.
5. In table 2, sort the parameters according to the type of power source.
6. The country of production of the research equipment should be added as required by the journal.
7. Is the name of the scanning microscope not changed by any chance? I think it's "SUPRA 55" and not "SUPARR 55". If I am wrong please ignore this remark.
8. In Figures 3a and c the arrows should point in the opposite direction (Unmelted Zn)
9. The References list should be extended, especially with new articles related to dissimilar joints using TIG and LBW methods.

Round 2

Reviewer 1 Report

The manuscript has been modified well according to the comments. But English language and style are fine/minor spell check required.

Reviewer 2 Report

The authors have corrected the paper according to the review amendments. Therefore the paper can be published in the provided version. 

Reviewer 3 Report

I have no more comments. The manuscript was sufficiently improved and therefore I can recommend acceptation in MDPI Materials.